# Pre-Training and Fine-Tuning Image Super-Resolution Models for Efficient Video Super-Resolution

## Abstract

In this paper, we propose a novel framework for adapting pre-trained image super-resolution (SR) models to tackle the challenging task of efficient video SR. This is achieved by freezing the pre-trained image SR model and fine-tuning it with the addition of several lightweight adapter modules. These adapters facilitate spatial and temporal learning, progressively equipping the image SR model with spatiotemporal reasoning capabilities for video SR. Also, these Adapters are compact and extendable, embedding only a few trainable parameters for each video dataset. Moreover, the parameters of the image SR model remain unchanged, resulting in substantial parameter sharing. This allows us to train video SR models quickly and efficiently. Remarkably, despite having significantly fewer parameters, our proposed method achieves competitive or even superior performance compared to existing video SR methods across multiple benchmarks.

## 1 Introduction

In recent years, super-resolution (SR) techniques, which aim to enhance the quality of images and videos by increasing their resolution, have become a hot topic in the field of computer vision. With the advent of deep learning, the development of SR models has been significantly accelerated, leading to impressive improvements in image and video quality (Chan et al., 2021; 2022a; Yang et al., 2021b; Chan et al., 2022b; Chu et al., 2020; Haris et al., 2020). However, three key obstacles emerge when training video SR models. Firstly, such models necessitate more computational resources and memory than their image SR counterparts, escalating the difficulty of their training and deployment. Secondly, the inherent high dimensionality of video data coupled with the intricate nature of video SR models can cause instability during the training process. Finally, the scarcity of high-quality video SR datasets compared to image ones poses a challenge in training models that effectively generalize across diverse video content.

One possible approach is to bootstrap an SR model pre-trained on images and then fine-tune it on video data. However, applying these sophisticated SR models to video sequences is not straightforward and introduces new challenges, including the need to deal with temporal dependencies and the complexity of motion information in video data. To this end, we propose a novel framework for efficient video SR that capitalizes on the power of pre-trained image SR models. Our approach termed Pre-training and Fine-tuning Video Super-Resolution (PFVSR), is designed to address the unique challenges posed by video data. We are motivated by the observation that pre-trained image SR models can provide a solid starting point for video SR, given they are appropriately adapted and fine-tuned to handle the intricacies of video data.

In the first phase of our method, the pre-training phase, we train an image SR model on a vast amount of image data, allowing it to learn spatial details that are crucial for image enhancement. Following this, we move into the fine-tuning phase, wherein we introduce a series of lightweight adapter modules (Houlsby et al., 2019) into the pre-trained image SR model. These adapters are designed to capture temporal information across video frames and integrate it with the spatial details learned in the pre-training phase. To be specific, we commence by incorporating an adapter module, as demonstrated in Figure 1b, following the self-attention layer in a Swin Transformer block (see Figure 1a). This facilitates spatial adaptation, as visualized in Figure 1c. We find that a well-pre-trained

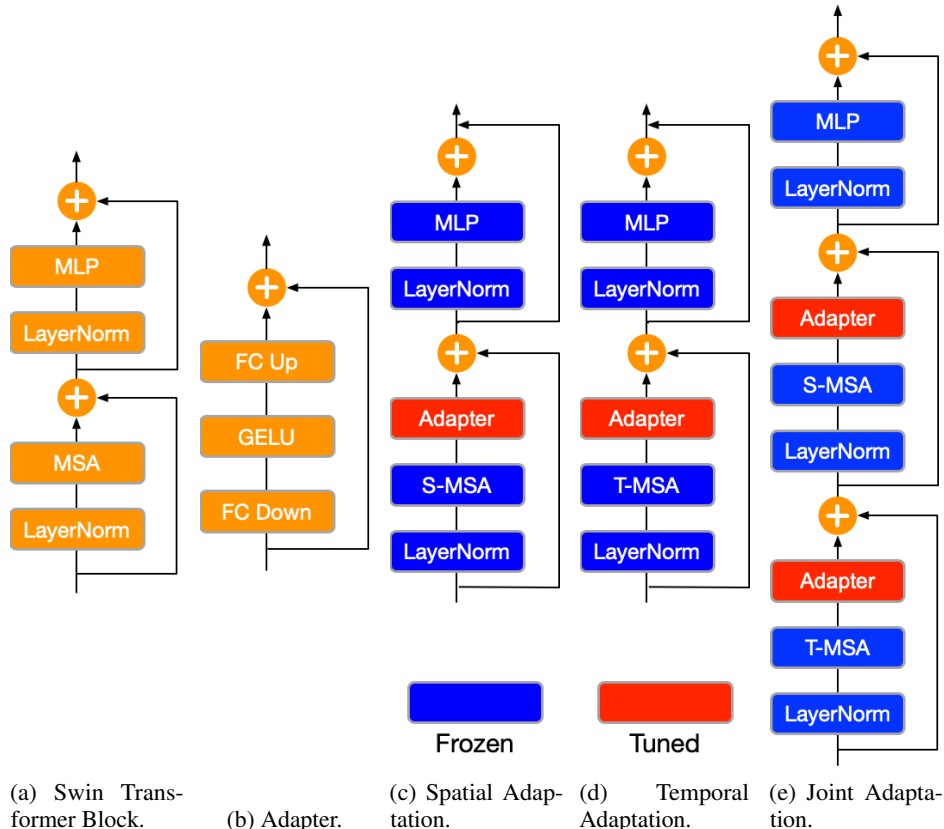

(a) Swin Transformer Block.  (b) Adapter.  (c) Spatial Adaptation.  (d) Temporal Adaptation.  (e) Joint Adaptation.

Figure 1: A detailed illustration of how we modify a conventional Swin Transformer block (a) to address the task of video SR by systematically incorporating spatial adaptation (c) and temporal adaptation (d). Our completed framework (e) integrates both these adaptations. It's imperative to note that while the S-MSA and T-MSA share weights, they operate on different input dimensions. Throughout the training process, only the newly incorporated Adapter (b) modules undergo updates (marked in red), while the rest of the layers remain in a frozen state (marked in blue). This approach dramatically reduces the parameter space that needs to be explored during training, leading to significant computational savings without compromising performance.

image model is highly effective for spatial modeling in video generation tasks. Subsequently, we turn our attention to temporal modeling. To this end, we retain the pre-trained self-attention layer from the image model but repurpose it for the temporal dimension of video input. This strategy enforces the model to establish correlations across different frames. An additional adapter is also implemented for temporal adaptation, as illustrated in Figure 1d. Ultimately, we carry out a joint adaptation process by incorporating both spatial and temporal adapters into a Swin Transformer block, as shown in Figure 1e. This procedure significantly enhances the model's capability to handle video SR tasks effectively and efficiently.

Through this two-step approach, PFVSR efficiently adapts a pre-trained image SR model to video SR tasks, enabling it to understand and reproduce the temporal dynamics in video sequences while enhancing spatial resolution. PFVSR takes advantage of the rich spatial feature representations learned from the image SR pre-training phase and extends it by learning temporal dependencies in the fine-tuning phase, offering a robust and efficient solution to video SR.

In extensive experiments, we demonstrate that PFVSR significantly enhances the efficiency of video SR without compromising the output quality. Notably, our method achieves much better performance compared to existing methods, despite having significantly fewer parameters and lower computational complexity. This improvement in efficiency makes PFVSR particularly suitable for real-world applications where both performance and computational efficiency are important considerations. We hope that our work will open up new avenues for the development of efficient and high-performance video SR frameworks. To summarize, we make the following contributions:

- We propose a new approach for adapting pre-trained image SR models to efficiently handle the video SR task. Our method is highly versatile and applicable to various pre-trained image SR models. It is straightforward to implement and offers cost-effective training benefits.
- Significantly, our method exhibits superior efficiency compared to existing video SR models. For instance, when juxtaposed with the current state-of-the-art video SR model, RVRT (Liang et al., 2022b), our approach delivers substantial performance improvements while utilizing at least 15% fewer model parameters, 20% less testing memory, and reducing runtime by 15%.
- We validate our approach through extensive experiments on several public datasets, where our method consistently delivers better results than existing methods. To further foster research, we will make the source code and models publicly available. This step ensures transparency and allows the scientific community to build upon our work, potentially leading to even more efficient and effective video SR models.

## 2 RELATED WORK

**Image Pre-Trained Models.** Vision Transformer (ViT) and its related variants, as introduced by Dosovitskiy et al. (Dosovitskiy et al., 2021), have played a pivotal role in breaking new ground on a wide array of computer vision tasks. This broad spectrum of tasks spans from image segmentation (Wang et al., 2021a;b; Jain et al., 2023), object detection (Carion et al., 2020; Zhu et al., 2021; Dai et al., 2022; Hassani et al., 2023), depth estimation (Yang et al., 2021a), and pose estimation (Li et al., 2022; Lin et al., 2021b), video inpainting (Zeng et al., 2020), vision-and-language navigation (Chen et al., 2021b), video classification (Neimark et al., 2021), 3D pose transfer (Chen et al., 2022; 2021a), and house layout generation (Tang et al., 2023). Once these models are trained, they establish a robust foundation that can be effectively transferred and applied to downstream tasks through fine-tuning (Zhai et al., 2022; Xie et al., 2022; Jia et al., 2021; 2022). For example, Jia et al. (Jia et al., 2022) presented visual prompt tuning (VPT), a method that offers a resource-efficient and highly effective alternative to the standard full fine-tuning approach typically used with large-scale Transformer models in the visual domain.

In this paper, we exploit the simplicity of our proposed method to harness the capabilities of these well-pre-trained image models and adapt them efficiently for video tasks. Specifically, we aim to utilize these adeptly pre-trained image SR models for efficient video SR tasks, thereby making a significant stride in the domain of video SR.

**Video Super-Resolution (VSR)** is a challenging task that aims to generate high-resolution videos from their lower-resolution versions. The primary difficulty in VSR lies in effectively leveraging the complementary details available in adjacent frames, which may often be misaligned due to movements within the scene or camera motion. Numerous existing VSR methods, including TDAN (Tian et al., 2020), EDVR (Wang et al., 2019), MuCAN (Li et al., 2020), DynaVSR (Lee et al., 2021), DSMC (Liu et al., 2021), OVSR (Yi et al., 2021), TMNet (Xu et al., 2021), FRVSR (Sajjadi et al., 2018), SPMC (Tao et al., 2017), RBPN (Haris et al., 2019), PFNL (Yi et al., 2019), TGA (Isobe et al., 2020b), BasicVSR (Chan et al., 2021), IconVSR (Chan et al., 2021), BasicVSR++(Chan et al., 2022a), RSDN(Isobe et al., 2020a), RLSP (Fuoli et al., 2019), DUF (Jo et al., 2018), and BRCN (Huang et al., 2015) have managed to generate satisfactory results through their carefully engineered VSR models. However, these models typically require training from scratch, resulting in considerable GPU resource consumption and training time.

In this paper, we introduce a novel strategy for repurposing pre-trained image SR models for VSR tasks. This novel approach, a first of its kind, enables us to simply fine-tune the pre-trained image SR models rather than starting from scratch. As a result, we substantially decrease the demand for GPU resources and training time, making our method far more efficient and practical.

**Parameter-Efficient Fine-Tuning** strategies have their roots in the realm of NLP. The growing complexities and size of language models, along with the need to adapt them to a plethora of downstream tasks, have led to the development of these strategies (He et al., 2022; Houlsby et al., 2019). The central goal of these methods is to minimize the number of trainable parameters, thereby reducing computational overhead while maintaining or even exceeding the performance achieved by complete fine-tuning. For instance, He et al. (He et al., 2022) introduced a unified framework that consolidates various effective parameter-tuning methods. This enables us to construct a more efficient model that matches the performance of full fine-tuning by cross-applying techniques from different

approaches. Houlsby et al. (Houlsby et al., 2019) proposed the concept of transfer with adapter modules, resulting in compact and easily extendable models. These models only add a minimal amount of trainable parameters per task, thereby enabling the incorporation of new tasks without the need to revisit previous ones. The parameters of the original network remain unaltered, resulting in a high degree of parameter sharing. Of late, this parameter-efficient fine-tuning concept has made its way into the computer vision domain (Yang et al., 2023; Lin et al., 2022). For instance, Lin et al. (Lin et al., 2022) introduced efficient video learning (EVL), which is a streamlined framework for directly training high-quality video recognition models using frozen CLIP features. Similarly, Yang et al. (Yang et al., 2023) proposed a novel method for adapting pre-trained image models (AIM) to video action recognition tasks. AIM has demonstrated performance that is comparable to, or even surpasses, previously fully fine-tuned state-of-the-art models on four video action recognition benchmarks.

While this technique has found applications in numerous computer vision tasks, its application to the field of video SR is a pioneering attempt. To the best of our knowledge, we are the first to propose adapting pre-trained image SR models to tackle the VSR task.

## 3 Pre-training and Fine-Tuning Video Super-Resolution

In this section, we commence our discussion with a concise overview of the Swin Transformer block, illuminating its primary architecture and functionalities. This will serve as a foundation for understanding the techniques we utilize in our proposed method. Next, we delve into the specifics of spatial adaptation. We demonstrate how we leverage this method to fine-tune a pre-trained image SR model to better understand and process spatial aspects of video data. Moving on, we introduce the concept of temporal adaptation. We illustrate how this technique is employed to imbue our model with an understanding of the temporal dynamics inherent in video data, thus enhancing its capability in the video SR task. Subsequently, we explore the process of joint adaptation, which is a harmonious combination of spatial and temporal adaptations. This stage represents the culmination of our adaptation process, where we integrate the knowledge gained from both spatial and temporal adaptations into our pre-trained image SR model. This integrated approach propels the model's performance, making it highly effective for the video SR task. Throughout this section, we aim to elucidate the step-by-step process of adapting an image SR model for the video SR task, offering a detailed insight into the effectiveness of our proposed method.

### 3.1 Swin Transformer Block

This paper focuses on the process of adapting pre-trained Swin Transformer image models to the video SR task and compares their performance with fully trained video SR Transformer models. We consider using Swin Transformer because it achieves good results in image SR (Liang et al., 2021). Figure 1a shows the Swin Transformer block's unique handling of inputs of size $H \times W \times C$. It reconfigures the input into a $\frac{HW}{M^2} \times M^2 \times C$ feature by breaking it down into non-overlapping $M \times M$ local windows. Within each window, the Swin Transformer computes self-attention. For each local window feature $F \in \mathbb{R}^{M^2 \times C}$, it calculates the matrices $Q$, $K$, and $V$ as:

$$Q = FP_Q, \quad K = FP_K, \quad V = FP_V, \tag{1}$$

with $P_Q$, $P_K$, and $P_V$ as shared projection matrices. The attention matrix is then derived as $\text{Attention}(Q, K, V) = \text{Softmax}(QK^T/\sqrt{d} + B)V$, with $B$ being the learnable relative positional encoding. The Swin Transformer also employs an MLP for further feature transformations. Both MSA and MLP are preceded by a LayerNorm (LN) layer, and residual connections are used in both cases. This process is summarized as:

$$F = MSA(LN(F)) + F, \quad F = MLP(LN(F)) + F. \tag{2}$$

To overcome the lack of connections between local windows when partitioning is consistent, the Swin Transformer alternates between regular and shifted window partitioning. The latter involves a pixel shift before partitioning, adding to the Transformer's flexibility and adaptability.

### 3.2 Spatial Adaptation for Video Super-Resolution

Pre-trained image models, trained on large-scale datasets, have shown exceptional transferability to numerous downstream computer vision tasks. Based on this strong performance, we hypothesize that

these models can be effectively fine-tuned to achieve high-quality spatial modeling in the domain of video super-resolution. This proposed approach is inspired by efficient fine-tuning techniques that have been successfully deployed in NLP (Houlsby et al., 2019; Li & Liang, 2021; Zaken et al., 2022). Among these techniques, we opt to implement Adapter (Houlsby et al., 2019), mainly due to their straightforward and intuitive architecture. As depicted in Figure 1b, the Adapter is a bottleneck structure composed of two fully connected (FC) layers, with an activation layer sandwiched in between. The primary role of the first FC layer is to project the input into a lower dimension, while the second FC layer reverses this operation, projecting it back to the original dimension. To tailor the pre-trained spatial features to target video data, we introduce an Adapter following the self-attention layer, as illustrated in Figure 1c. We refer to this as spatial adaptation. During the training phase, all other layers of the Swin Transformer block remain frozen, with only the Adapter being updated.

The effectiveness of the spatial adaptation strategy is demonstrated in Table 3 and Figure 3. We see from Table 3 that it significantly outperforms the pre-trained image SR baseline. These results suggest that spatial adaptation allows the frozen image SR model to learn robust spatial representations from video data. However, it is important to note that there still exists a considerable performance gap between spatial adaptation and a fully trained video SR model. This can primarily be attributed to the fact that spatial adaptation alone does not possess the capacity to learn temporal information inherent in videos. Thus, to bridge this gap, temporal adaptation becomes an indispensable component in our framework. This not only complements the spatial adaptation by allowing the model to learn and understand the temporal dynamics in video sequences but also enhances the overall performance of the VSR task. Through the combination of spatial and temporal adaptation, our approach aims to harness the strengths of both, creating a more comprehensive and effective solution for VSR.

### 3.3 TEMPORAL ADAPTATION FOR VIDEO SUPER-RESOLUTION

In order to effectively capture temporal information in videos for video SR, we propose a novel strategy: reusing the pre-trained self-attention layer from the image SR model for temporal modeling. More specifically, we designate the original self-attention layer as S-MSA for spatial modeling and the repurposed self-attention layer as T-MSA for temporal modeling. As illustrated in Figure 1e, we position T-MSA ahead of S-MSA. Given the video patch embedding $v \in \mathbb{R}^{T \times (N+1) \times D}$, our initial step is to reshape it into $v^T \in \mathbb{R}^{(N+1) \times T \times D}$, where $N = HW/P^2$ is the number of spatial patches, $P$ denotes the patch size, and $T$ is the number of frames. We then feed $v^T$ into the T-MSA where it endeavors to learn the relationship among the $T$ frames. It's important to note that T-MSA and S-MSA are the same layers (i.e., the pre-trained MSA in the image SR model) and remain frozen during model tuning but are applied to different input dimensions. This explicit operation enhances the model's temporal modeling capability without increasing the number of parameters. Following the same principle as spatial adaptation, we incorporate another Adapter after the repurposed temporal attention layer to adapt its features to video data. This is referred to as temporal adaptation (Figure 1d). The Adapter's structure is identical to that in spatial adaptation. As evidenced by the results in Table 3, temporal adaptation successfully narrows the gap to fully trained video SR models, while only introducing another lightweight Adapter into the Swin Transformer block.

Despite these encouraging results, our straightforward strategy of reusing spatial attention for temporal modeling may not be sufficiently robust for video SR with complex temporal dynamics. To counteract this, we integrate a new temporal module into the pre-trained image SR models, given the common understanding that image models may struggle to infer temporal structured information in videos. Specifically, we adopt the trajectory-aware attention (Liu et al., 2022) to capture intricate temporal information. Although this method increases the number of tunable parameters of the model, it significantly enhances the model's performance, as confirmed by the results in Table 3. This demonstrates the value of specifically designed temporal modules in improving video super-resolution performance, especially for challenging videos with complex temporal structures.

### 3.4 JOINT ADAPTATION FOR VIDEO SUPER-RESOLUTION

Spatial and temporal adaptations are carried out sequentially, each focusing on distinct input dimensions and serving unique roles. Spatial adaptation primarily focuses on adapting pre-trained image features to the video context, while temporal adaptation aims to instill temporal dynamics into the model. This process effectively fine-tunes the video representations for comprehensive spatiotemporal

reasoning, as illustrated in Figure 1e. The sequential nature of this process ensures that each step is focused and purposeful. The spatial adaptation step serves as a foundation, adapting the pre-trained model to handle the spatial characteristics of video data. Subsequently, the temporal adaptation step builds on this foundation, incorporating the crucial temporal dimension that is inherent in video data. This stepwise procedure ensures that the model gradually acquires the necessary skills for video super-resolution, without overwhelming the learning process.

This structured approach to adaptation not only enhances the model's performance on video SR tasks but also exhibits the potential to be easily extended and adapted for other video-related tasks. By isolating spatial and temporal adaptations, it becomes easier to experiment with different strategies and modules for each component, potentially leading to further improvements in performance.

## 4 EXPERIMENTS

### 4.1 EXPERIMENTAL SETTINGS

**Datasets.** In this paper, we align with the approach taken by RVRT (Liang et al., 2022b) and concentrate our efforts on two specific degradation scenarios: bicubic (BI) and blur-downsampling (BD). Both of these scenarios involve an upscaling factor of $\times 4$, demanding the model to magnify the input data by four times. For BI degradation, we make use of two distinct datasets to train our model. The first is the REDS dataset (Nah et al., 2019), and the second is the Vimeo-90K dataset (Xue et al., 2019). Each dataset has been carefully chosen, offering a diverse range of characteristics to help fine-tune our model. Following the training phase, we proceed to evaluate our model's performance using the corresponding test subsets of these datasets, namely REDS4 and Vimeo-90K-T. The REDS4 test subset consists of specific clips numbered 000, 011, 015, and 020, offering a robust test of our model's capabilities. We complement these tests by introducing an additional dataset, Vid4 (Liu & Sun, 2013), alongside Vimeo-90K for further validation of our model's performance. Regarding BD degradation, we employ the Vimeo-90K dataset as the training set for our model. This dataset provides a comprehensive range of blur-downsampling examples that allow us to fine-tune our model effectively. Following the training, we assess our model on three test datasets: Vimeo-90K-T, Vid4, and UDM10 (Yi et al., 2019). These datasets present varying levels of challenge and complexity, ensuring our model's performance is thoroughly evaluated under diverse BD degradation conditions.

**Implementation Details.** In this paper, we detail our proposed two-stage training process, which begins with pre-training on an image dataset and concludes with fine-tuning on a video dataset. More specifically, during the first stage, we follow the training approach outlined in SwinIR (Liang et al., 2021) to pre-train our model on the DIV2K (Lim et al., 2017) + Flickr2K (Timofte et al., 2017) dataset. Subsequently, in the second stage, we implement the training strategy from RVRT (Liang et al., 2022b) to fine-tune the model on specific video datasets, such as REDS. The proposed strategy, referred to as "pre-training and fine-tuning", lies in its simplicity and capability to yield significant performance improvements. We believe that the effectiveness of this approach greatly depends on a sufficient number of training iterations during the pre-training phase and an appropriately small learning rate during the fine-tuning phase. This is due to the nature of the Transformer, which requires extensive data and iteration cycles to acquire a generalized understanding of the task, yet necessitates a small learning rate during fine-tuning to prevent overfitting to the specific video dataset.

For fine-tuning training, we emulate the training procedure established by RVRT (Liang et al., 2022b). The model is trained for 300,000 iterations using the Adam optimizer (Kingma & Ba, 2015) with default settings and a batch size of 8. Notably, RVRT requires 600,000 iterations for training, while our method achieves better results in just 300,000 iterations, showcasing its superior training efficiency. The learning rate is initially set at $4e^{-4}$ and gradually decreased in accordance with the Cosine Annealing scheme (Loshchilov & Hutter, 2017). To ensure stable training, we follow RVRT and Basicvsr++, and initialize the SpyNet (Ranjan & Black, 2017) with pre-trained weights, maintain it in a fixed state for the initial 20,000 iterations, and subsequently reduce its learning rate by 75%.

### 4.2 EXPERIMENTAL RESULTS

**State-of-the-Art Comparisons.** In our experiments, we position our proposed method, PFVSR, in a highly competitive landscape, pitting it against 19 of the most notable SOTA approaches in VSR, as shown in Table 1. We opt for this extensive list of methods to ensure a comprehensive and thorough

Table 1: State-of-the-art comparison (PSNR/SSIM). All results are calculated on Y-channel except REDS4 (RGB-channel).

| Method | BI Degradation | | | BD Degradation | | |
|---|---|---|---|---|---|---|
| | REDS4 | Vimeo-90K-T | Vid4 | UDM10 | Vimeo-90K-T | Vid4 |
| Bicubic | 26.14/0.7292 | 31.32/0.8684 | 23.78/0.6347 | 28.47/0.8253 | 31.30/0.8687 | 21.80/0.5246 |
| TOFlow (Xue et al., 2019) | 27.98/0.7990 | 33.08/0.9054 | 25.89/0.7651 | 36.26/0.9438 | 34.62/0.9212 | 25.85/0.7659 |
| FRVSR (Sajjadi et al., 2018) | - | - | - | 37.09/0.9522 | 35.64/0.9319 | 26.69/0.8103 |
| DUF (Jo et al., 2018) | 28.63/0.8251 | - | 27.33/0.8319 | 38.48/0.9605 | 36.87/0.9447 | 27.38/0.8329 |
| PFNL (Yi et al., 2019) | 29.63/0.8502 | 36.14/0.9363 | 26.73/0.8029 | 38.74/0.9627 | - | 27.16/0.8355 |
| RBPN (Haris et al., 2019) | 30.09/0.8590 | 37.07/0.9435 | 27.12/0.8180 | 38.66/0.9596 | 37.20/0.9458 | 27.17/0.8205 |
| MuCAN (Li et al., 2020) | 30.88/0.8750 | 37.32/0.9465 | - | - | - | - |
| RLSP (Fuoli et al., 2019) | - | - | - | 38.48/0.9606 | 36.49/0.9403 | 27.48/0.8388 |
| TGA (Isobe et al., 2020b) | - | - | - | 38.74/0.9627 | 37.59/0.9516 | 27.63/0.8423 |
| RSDN (Isobe et al., 2020a) | - | - | - | 39.35/0.9653 | 37.23/0.9471 | 27.92/0.8505 |
| RRN (Isobe et al., 2020c) | - | - | - | 38.96/0.9644 | - | 27.69/0.8488 |
| FDAN (Lin et al., 2021a) | - | - | - | 39.91/0.9686 | 37.75/0.9522 | 27.88/0.8508 |
| EDVR (Wang et al., 2019) | 31.09/0.8800 | 37.61/0.9489 | 27.35/0.8264 | 39.89/0.9686 | 37.81/0.9523 | 27.85/0.8503 |
| GOVSR (Yi et al., 2021) | - | - | - | 40.14/0.9713 | 37.63/0.9503 | 28.41/0.8724 |
| VSRT (Cao et al., 2021) | 31.19/0.8815 | 37.71/0.9494 | 27.36/0.8258 | - | - | - |
| BasicVSR (Chan et al., 2021) | 31.42/0.8909 | 37.18/0.9450 | 27.24/0.8251 | 39.96/0.9694 | 37.53/0.9498 | 27.96/0.8553 |
| IconVSR (Chan et al., 2021) | 31.67/0.8948 | 37.47/0.9476 | 27.39/0.8279 | 40.03/0.9694 | 37.84/0.9524 | 28.04/0.8570 |
| VRT (Liang et al., 2022a) | 32.19/0.9006 | 38.20/0.9530 | 27.93/0.8425 | 41.05/0.9737 | 38.72/0.9584 | 29.42/0.8795 |
| PSRT (Shi et al., 2022) | 32.72/0.9106 | 38.27/0.9536 | 28.07/0.8485 | -/- | -/- | -/- |
| BasicVSR++ (Chan et al., 2022a) | 32.39/0.9069 | 37.79/0.9500 | 27.79/0.8400 | 40.72/0.9722 | 38.21/0.9550 | 29.04/0.8753 |
| RVRT (Liang et al., 2022b) | 32.75/0.9113 | 38.15/0.9527 | 27.99/0.8462 | 40.90/0.9729 | 38.59/0.9576 | 29.54/0.8810 |
| PFVSR (Ours) | 32.87/0.9135 | 38.24/0.9533 | 28.05/0.8467 | 40.96/0.9734 | 38.64/0.9581 | 29.58/0.8817 |
| PFVSR2 (Ours) | **33.08/0.9172** | **38.37/0.9586** | **28.23/0.8502** | **41.28/0.9756** | **38.74/0.9597** | **29.71/0.8848** |
| PFVSR3 (Ours) | 32.90/0.9148 | 38.26/0.9552 | 28.18/0.8483 | 41.14/0.9740 | 38.63/0.9585 | 29.62/0.8829 |

evaluation, pushing our method to its limits and assessing its performance in a variety of contexts. The quantitative results of these head-to-head comparisons are concisely presented in Table 1. Our PFVSR either matches or surpasses the performance of existing SOTA methods in terms of PSNR and SSIM metrics across two different degradation conditions, thereby underscoring the effectiveness of our approach and positioning it as a promising candidate for future developments and applications in the realm of VSR. To further improve the performance of our method, we continue to train our model for 600,000 iterations (PFVSR2) and achieve better results, which are significantly better than the results of RVRT (0.25db on average). In addition, although it is also training 600,000 iterations, under the same GPU conditions. RVRT takes about 51 hours, while our method only takes 27 hours. This demonstrates the high efficiency of our method.

Furthermore, as depicted in Figure 2, our method, PFVSR, does more than just generate visually appealing results; it excels in preserving the intricate textures and details that contribute to the VSR, where the objective is not only to enhance the resolution but also to maintain the authenticity of the original content. Remarkably, PFVSR outperforms other leading approaches in this aspect, including EDVR, BasicVSR, BasicVSR++, VRT, and RVRT. These methods, while formidable in their own right, do not achieve the same level of detail preservation that our method does. This accomplishment underlines the effectiveness of our approach and its potential to pave the way for future advancements in VSR techniques.

**Data Efficiency.** Our method requires less video data. In order to validate this idea, we only used 60% of the data of each video dataset for training. After the same training of 600,000 iterations, our method still achieved better results than RVRT, as shown in the PFVSR3 results in Table 1.

**Model Efficiency.** We undertake a comprehensive comparison of various models focusing on model size, memory consumption during testing, and runtime. The results are listed in Table 2. Notably, PFVSR stands out among the representative parallel methods, which include EDVR, VSRT, VRT, and RVRT. PFVSR manifests significant performance improvements, all the while utilizing fewer resources. Specifically, it uses at least 15% fewer model parameters and

Table 2: Model size, testing memory, and runtime (ms) comparison for a low-resolution of $320 \times 180$. Our PFVSR could serve as a good candidate for VSR when training resources are more limited.

| Method | # Params | Memory | Runtime | PNSR ↑ |
|---|---|---|---|---|
| BasicVSR++ (Chan et al., 2022a) | 7.3M | 223M | 77 | 32.39 |
| EDVR (Wang et al., 2019) | 20.6M | 3535M | 378 | 31.09 |
| VSRT (Cao et al., 2021) | 32.6M | 27487M | 328 | 31.19 |
| VRT (Liang et al., 2022a) | 35.6M | 2149M | 243 | 32.19 |
| RVRT (Liang et al., 2022b) | 10.8M | 1056M | 183 | 32.75 |
| PFVSR (Ours) | 9.1M | 843M | 152 | **32.87** |

requires 20% less memory during testing. Furthermore, the runtime of PFVSR is trimmed by a minimum of 15% when compared with these parallel methods, offering a more efficient alternative. When pitted against the recurrent model, BasicVSR++, PFVSR presents an impressive improvement

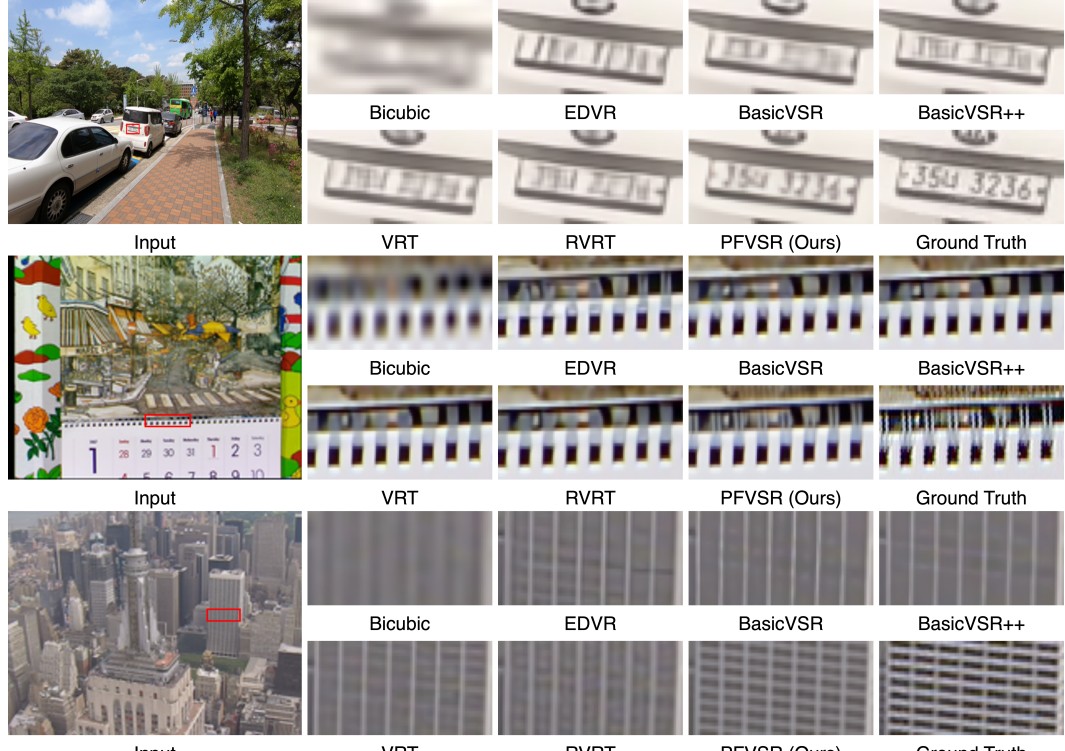

Figure 2: Visualization comparisons with existing state-of-the-art VSR methods, i.e., EDVR (Wang et al., 2019), BasicVSR (Chan et al., 2021), BasicVSR++ (Chan et al., 2022a), VRT (Liang et al., 2022a), and RVRT (Liang et al., 2022b). Zoom in for the best view.

in performance. It registers a PSNR increment of 0.48dB, marking a noteworthy advancement. This comparison underscores the effectiveness of our proposed PFVSR model, which offers an optimal balance of performance, efficiency, and resource utilization.

We also include the number of parameters, memory, and runtime of each proposed module in Table 3. This makes it clearer how each module affects the overall performance of the model. Note that only the spatial adapter and temporal adapter contain learnable network parameters. The parameters of the spatial adapter and temporal adapter are 3.2M and 5.9M, respectively. Therefore, the parameters of B1, B2, B3, B4, and B5 are 0M, 3.2M, 9.1M, 9.1M, and 9.1M, respectively.

### 4.3 Ablation Study

We conduct extensive ablation studies on REDS to evaluate each component of the proposed method.

**Baseline Models.** We introduce and evaluate five variants (namely, B1, B2, B3, B4, and B5), the specifics of which are outlined in Table 3. To elaborate, (1) our first baseline, B1, employs the pre-trained SwinIR model to conduct tests directly on the video dataset, acting as our fundamental evaluation point. (2) Our second baseline, B2, builds upon the foundation of B1 by integrating the spatial adaptation technique as portrayed in Figure 1c, thereby commencing the process of model fine-tuning for the task at hand. (3) Proceeding further, B3 extends the model of B2 by integrating the temporal adaptation strategy as proposed in Figure 1d. This inclusion enhances the model's capacity to comprehend and utilize temporal information from the video sequences. (4) In the case of B4, we chose to replace the temporal adaptation method integrated in B3 with the Trajectory-aware Attention mechanism as proposed by (Liu et al., 2022). This adjustment was aimed at comparing the relative effectiveness of different temporal adaptation methods. (5) Lastly, B5 represents our finalized model. In an effort to further bolster performance, we substitute the backbone of the B4 model with a HAT network as proposed by (Chen et al., 2023). This final change completes our model's evolution, yielding a superior solution that efficiently addresses the video super-resolution problem.

Table 3: The ablation study of the proposed PFVSR on REDS4.

| # | Method | PSNR ↑ | SSIM ↑ | # Params | Memory | Runtime |
|---|---|---|---|---|---|---|
| B1 | SwinIR (Liang et al., 2021) | 29.13 | 0.8272 | 0M | 165M | 54 |
| B2 | B1 + Spatial Adaptation (Fiure 1c) | 30.25 | 0.8531 | 3.2M | 287M | 78 |
| B3 | B2 + Temporal Adaptation (Fiure 1d) | 31.98 | 0.8979 | 9.1M | 603M | 116 |
| B4 | B2 + Temporal Adaptation (Trajectory-aware Attention (Liu et al., 2022)) | 32.46 | 0.9056 | 9.1M | 768M | 145 |
| B5 | B4 → HAT Backbone (Chen et al., 2023) | **32.87** | **0.9135** | 9.1M | 843M | 152 |

**Effect of Spatial and Temporal Adaptation.** The goal of our approach is to introduce a minimal number of tunable parameters to the frozen image SR model, thereby bridging the performance disparity with fully trained video SR models. As reflected in Table 3, the introduction of spatial adaptation in B2 leads to a substantial performance improvement over B1. This demonstrates that spatial adaptation plays a crucial role in enabling frozen image SR models to excel at spatial modeling tasks in video SR. Further, the integration of temporal adaptation in B3 provides an additional boost in performance. This enhancement validates the potency of our temporal adaptation strategy, proving that it can effectively impart robust temporal modeling capabilities to a model originally designed for spatial-only models. These findings collectively underscore the efficacy of the proposed spatial and temporal adaptation strategies. They suggest that by making calculated, incremental changes to a pre-trained image SR model, we can remarkably enhance its performance in the video SR domain without necessitating a complete retraining process.

**Effect of Different Temporal Adaptation Strategies.** Even though our straightforward strategy of reusing spatial attention to temporal modeling yields encouraging outcomes, it might not be adequately effective for videos with demanding temporal intricacies. Temporal modeling in videos can be treated as a type of sequence modeling, which led us to substitute the temporal adaptation method in B3 with the trajectory-aware attention (Liu et al., 2022). This attention mechanism integrates relevant visual tokens existing in identical spatiotemporal trajectories, thereby leading to enhanced performance and reduced computational demands. It is observed that B4 outperforms B3 on both evaluation metrics, validating that the independent design of the temporal adaptation module can bring about substantial performance improvements. Importantly, we have the flexibility to utilize an existing temporal modeling module to further optimize performance, such as temporal attention (Bertasius et al., 2021) or temporal encoder/decoder (Lin et al., 2022).

**Effect of Different Pre-Trained Image SR Models.** The elegance of our approach lies in its simplicity and universality, making it adaptable to more sophisticated image SR models. In order to substantiate this claim, we switch the SwinIR image model in B4 with a more potent HAT backbone (Chen et al., 2023). The resultant B5 outperforms B4, thereby reinforcing our foundational motivation. Moreover, this experiment underscores the flexibility and extensibility of our approach, demonstrating its potential to be integrated with future advances in image SR models, potentially leading to further breakthroughs in the field of VSR.

We provide more analysis of experimental results in the Appendix.

## 5  CONCLUSION AND LIMITATIONS

In this paper, we introduce a novel framework (i.e., PFVSR) that effectively leverages pre-trained image SR models for the task of efficient video SR. This is accomplished by sequentially implementing spatial learning and temporal learning to incrementally instill spatiotemporal reasoning capabilities into the pre-trained image SR model. Notably, our approach only requires updates to the newly incorporated adapters modules, leading to substantial reductions in training costs compared to existing video SR models. Despite this cost efficiency, our method demonstrates performance that is on par with or surpasses that of existing state-of-the-art models across multiple benchmarks.

It is worth noting that we are the first to propose adapting pre-trained image SR models for efficient video SR tasks. This is not a trivial task, requiring many key modifications to existing models to make the proposed framework work. Moreover, our method is generally applicable to different image pre-trained SR models, simple to implement, and cost-effective to train. We believe that this paper makes an important step towards efficient video SR tasks. While our current model solely utilizes image modality for VSR, a potential area for future enhancement could involve integrating pre-trained models from text or audio domains alongside images to address this challenging VSR task.

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

## A  APPENDIX

### A.1  THE ORDER BETWEEN ADAPTERS AND ATTENTION BLOCKS

As we can see in Figure 1, is it better for the adapter to follow the attention block (S-MSA or T-MSA). In the initial experiment, we also tried to put both adapters in front of both attention blocks (S-MSA and T-MSA), but we found that the result was not as good as the adapters behind the attention blocks. The results are compared in Table 4.

Table 4: The ablation study of the order between T-MSA and S-MSA.

| Method | PSNR ↑ | SSIM ↑ |
|---|---|---|
| Attention Blocks to follow Adapters | 31.32 | 0.8805 |
| Adapters to follow Attention Blocks | 31.98 | 0.8979 |

The superior performance of having the adapter follow the attention block compared to the opposite configuration can be attributed to the inherent workings of both components. When the adapter follows the attention block, it refines and adjusts the feature representations that have already undergone attention-based transformation. In other words, the attention mechanism first helps in capturing long-range dependencies and important contextual information from the input. The subsequent adapter then refines these attention-processed features, making them more suitable for the specific task at hand. On the contrary, when the attention block follows the adapter, the adapter might introduce specific task-oriented biases or modifications to the features. The subsequent attention mechanism, which is designed to capture global context, might then operate on these biased features, leading to less optimal performance. Furthermore, attention mechanisms can be viewed as providing a broad overview or context, and having an adapter refine this context thereafter seems to be a more logical and effective flow for information processing in our experiments.

### A.2  THE ORDER BETWEEN T-MSA AND S-MSA

As shown in Figure 1e, we place T-MSA before S-MSA. The reason is that temporal information (T-MSA) might be more foundational to our method's operations, serving as a primary context. Once this context is established, the S-MSA can refine features with a richer temporal context in mind. Moreover, in our preliminary experiments, placing T-MSA before S-MSA showed an improvement in performance metrics compared to the reverse configuration, as shown in Table 5.

### A.3  EFFECT OF SPATIAL ADAPTATION

We also provide visual examples of adding spatial adaptation in Figure 3. We see that by using our proposed spatial adapter, we can get more details, improving the output quality. That means it is intended to allow the model to adjust to variations in spatial features, which might be complementary

Table 5: The ablation study of the order between T-MSA and S-MSA.

| Method | PSNR ↑ | SSIM ↑ |
|---|---|---|
| S-MSA before S-MSA | 31.64 | 0.8891 |
| T-MSA before S-MSA | 31.98 | 0.8979 |

to the temporal aspects. This means spatial adaptation is able to help frozen image SR models achieve good spatial modeling on video data. It's a perspective that further emphasizes the interconnectedness of spatial and temporal factors in the overall framework.

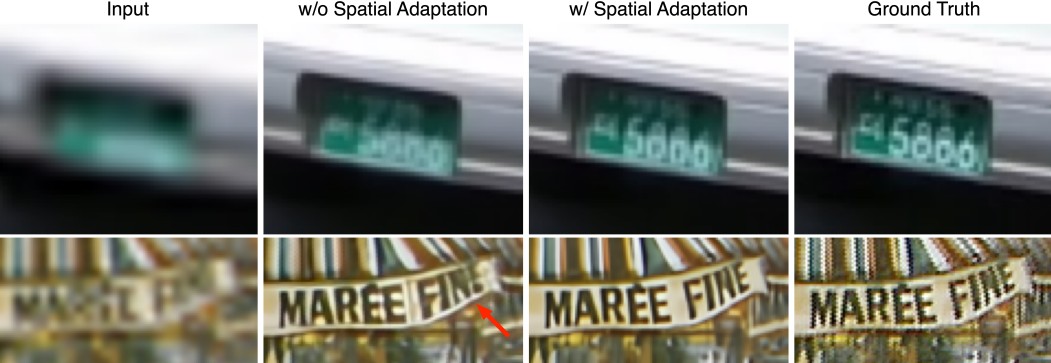

Figure 3: Analysis of the proposed Spatial Adaptation. By using the proposed spatial adapter, our framework leads to more details, improving the output quality.

## A.4 HOW PRE-TRAINING DATA AFFECT THE PERFORMANCE?

The amount and quality of data used in the pre-training phase can have a significant impact on the performance of the model. In the pre-training stage, the quality, diversity, and quantity of data can directly affect the generalization ability of the model and its subsequent fine-tuning performance. In the initial experiment, we only used DIV2K for pre-training and found that the result improvement was not obvious, and then we trained the model on a larger dataset (DIV2K+Flickr2K), the performance further improved significantly (up to 0.71db). However, this phenomenon is not caused by our proposed model, it is because Transformer-based models require more data to learn general knowledge for the task.

## A.5 DIFFERENT DEGRADATION COMBINATIONS

Since there are two degradations (BI and BD), we also provide discussions on different combinations, for example, BI (image model) + BD video fine-tuning. We conduct experiments with different degradation combinations on Vimeo-90K-T, the results are shown in Table 6.

Table 6: The ablation study of different degradation combinations.

| Method | PSNR ↑ | SSIM ↑ |
|---|---|---|
| BD (image model) + BI (video fine-tuning) | 37.51 | 0.9412 |
| BI (image model) + BI (video fine-tuning) | 38.24 | 0.9533 |
| BI (image model) + BD (video fine-tuning) | 37.89 | 0.9428 |
| BD (image model) + BD (video fine-tuning) | 38.64 | 0.9581 |

We can see that identical degradation combinations (e.g., BI+BI or BD+BD) perform better than mixed combinations (e.g., BI+BD or BD+BI), here are potential reasons: When a model is trained and fine-tuned within the same degradation setting, it might more effectively capture patterns and characteristics associated with that degradation. This can allow the model to adapt and optimize its performance more readily. Moreover, in a similar degradation environment, the updates to weights and biases via gradients might be more continuous and smooth, fostering more stable learning. Lastly, different degradations might introduce varying noise and distractions in the feature space. Using the

same degradation combination can reduce this variability, making it easier for the model to capture primary degradation patterns.

## A.6 TRAINING THE SAME NETWORK FROM SCRATCH USING VIDEO DATA

We train the same network from scratch using video data for 600,000 iterations, and the results are 32.81/0.9133, which is much worse than the proposed method (33.08/0.9172). We analyze the possible reason that our two-stage training method can make Transformer have a better ability to capture various variations and patterns present in input image and video data, enhancing its generalization ability. Moreover, training the same network from scratch using video data for 600,000 iterations needs about 59 hours, much slower than our proposed method (42 hours). These results fully prove the necessity of using pre-training and fine-tuning image super-resolution models to do the task of efficient video SR.

## A.7 EXTENTION TO CONVOLUTION-BASED IMAGE SR MODELS

The proposed adapter cannot be directly applied to convolution-based image SR models. The reason lies in the intrinsic structural and operational differences between Transformer-based architectures and convolution-based networks. Transformer and convolution-based models have distinct characteristics. Transformers tokenize and sequentialize the input data, which is vital for their self-attention to capture global dependencies. While convolution-based networks operate on grid-like data, like images, without the need for tokenization and sequentialization. This means that adapting Transformer-based adapters to convolution-based networks isn't straightforward, as the underlying data representation and processing paradigms differ fundamentally.

Addressing this challenge, we undertook a redesign process specifically for convolution-based networks. To make the adapter compatible, we made targeted modifications. In Figure 1b, we replaced the FC Down and FC Up with $1 \times 1$ convolutions and substituted GELU with ReLU. This yielded an adapter for convolution-based models. To validate its effectiveness, we integrated it into BSRGAN (Zhang et al., 2021), a well-established convolution-based architecture known for its strong performance in image SR. BSRGAN, which builds upon ESRGAN's structure (Wang et al., 2018), includes residual blocks. Our approach involved embedding the proposed convolution-based adapter after each residual block, facilitating effective domain adaptation and transfer.

In our experiments, the performance of the adapter was evaluated on the BI Degradation using REDS4. Our method achieved a PSNR/SSIM score of 32.53/0.9098, surpassing SOTA convolution-based VSR model, BasicVSR++ (32.39/0.9069). At the same time, the Runtime of our method in inference is much shorter than that of BasicVSR++ (62ms vs 77ms). These results validate the applicability of the convolution-based adapter and its ability to enhance convolution-based networks' performance. While the convolution-based adapter shows remarkable progress, the results still lag behind those of the Transformer-based method proposed in our paper. This difference could be attributed to Transformers' larger data demands compared to convolution-based approaches. The initial image pre-training stage in the Transformer-based method likely contributes to a more pronounced performance boost during fine-tuning, which may not be as pronounced in convolution-based networks.

Overall, our proposed Pre-Training and Fine-Tuning framework is applicable to both the Transformer-based and convolution-based image SR models.

## A.8 USER STUDY

We perform a user study to draw a comparison between our proposed method and several other leading methods, including the real video sequence. For the purpose of this study, we made a random selection of four video samples for each comparison. A total of ten participants took part in this study. They were tasked with responding to two specific questions pertaining to each video they reviewed. The first question (Q1) seeks to understand their perception of realism in the video, asking "Which video is more realistic?". The second question (Q2) aims to gauge the coherency of the video, asking "Which video is more coherent?". Both questions are designed to probe the users' subjective perceptions of realism and coherency in the videos, key aspects that contribute significantly to the overall quality of the video. The responses are presented in Table 7 as preference percentages, indicating the proportion of users who favored the results produced by each corresponding method or

the real video. The findings from this user study underscore the high quality of the videos generated by our proposed method. They substantiate our claim of superior realism and coherence, reinforcing our method's effectiveness and its potential for wider application in the field of video generation.

Table 7: User study results (%) (Q1/Q2).

| Method | BI Degradation | | | BD Degradation | | |
|---|---|---|---|---|---|---|
| | REDS4 | Vimeo-90K-T | Vid4 | UDM10 | Vimeo-90K-T | Vid4 |
| Bicubic | 0.5/1.3 | 0.7/2.2 | 0.8/1.5 | 0.4/1.2 | 0.6/1.9 | 0.7/2.1 |
| EDVR (Wang et al., 2019) | 5.4/3.2 | 4.1/3.5 | 3.2/2.8 | 3.8/3.2 | 4.7/3.5 | 3.9/4.2 |
| BasicVSR (Chan et al., 2021) | 8.9/7.3 | 6.4/5.7 | 6.8/4.7 | 7.4/6.6 | 8.9/6.4 | 5.7/7.3 |
| BasicVSR++ (Chan et al., 2022a) | 10.4/8.5 | 10.8/11.2 | 8.9/7.2 | 11.9/9.3 | 12.4/10.8 | 9.2/9.8 |
| VRT (Liang et al., 2022a) | 11.7/11.3 | 13.2/13.9 | 12.5/13.6 | 13.4/12.7 | 13.5/13.3 | 12.4/11.7 |
| RVRT (Liang et al., 2022b) | 13.6/13.7 | 16.5/14.3 | 15.7/16.0 | 15.2/16.8 | 14.4/15.1 | 14.1/14.8 |
| PFVSR (Ours) | **15.4/16.9** | **18.2/19.4** | **19.3/20.1** | **16.7/18.9** | **15.6/17.6** | **18.4/17.3** |
| Real Video | 34.1/37.8 | 30.1/29.8 | 32.8/34.1 | 31.2/31.3 | 29.9/31.4 | 35.6/32.8 |

## A.9 VIDEO DEMO

We provide a video demo for SOTA comparisons in the attached Supplementary Material. In the demo, we compare the proposed method with two SOTA methods, i.e., BasicVSR++ and RVRT. It can be seen that our method can generate sharper details, such as license plates and building windows.

