# OpenReview forum: "Pre-Training and Fine-Tuning Image Super-Resolution Models for Efficient Video Super-Resolution"
_ICLR.cc/2024/Conference — ICLR 2024 Conference Withdrawn Submission_

### Official Review · Reviewer_dM9L · 2023-10-18

**Soundness:** 3 good
**Presentation:** 2 fair
**Contribution:** 2 fair
**Rating:** 5
**Confidence:** 4

**Summary:**

This paper proposes an efficient video super-resolution (VSR) method based on image super-resolution(SR) models. It first pre-trains an exsiting image SR model on the image SR dataset. After pre-training, it freezes the parameters of image SR model and fine-tunes with several spatial / temporal adapters on video SR dataset. This method has fewer parameters while achieves competitive or even superior performance compared to existing video SR methods.

**Strengths:**

1. This paper shows that pre-trained parameters of an image SR model may be benefit for video SR.
2. It shows appealing performance against latest state-of-the-art video SR mothods.

**Weaknesses:**

1. Several implementation details are not clear.
(1) As shown in Fig. 1, there is lack of illustration on how to apply proposed model to video data. The model uses flow-based alignment (and maybe recurrent design) but is omitted in the figure.
(2) The paper demonstrates to use SwinIR as image SR model. However, from Tab. 3 in ablation study, I found it replaces with more powerful HAT backbone. Please make these details clear.
2. The comparison is unfaired.
(1) In Tab. 2, the #. Params of proposed PFVSR only considers the parameters of adapters, but without the parameters of backbone. The overall parameter count of PFVSR may be larger than RVRT.
(2) The training time comparion with RVRT do not consider the pre-training time of proposed method, which is also unfaired.
3. The adapters are inserted into each layer of image SR model, and contains 9.1M #. Params in total. They are not 'lightweight' adapters.

**Questions:**

In the reviewer's opinion, proposed PFVSR has comparable #. Params and runtime, etc. An amount of the parameters are pre-trained with image SR data, so the fine-tuned model shows better performance than state-of-the-art methods. FRVSR do not show much superiority in effciency. Besides, the written could be improved.

My questions are illustrated in Weaknesses.

---

### Official Review · Reviewer_sGmU · 2023-10-28

**Soundness:** 3 good
**Presentation:** 1 poor
**Contribution:** 2 fair
**Rating:** 3
**Confidence:** 5

**Summary:**

This paper proposed a new Transformer-based building block for the video super-resolution.
The block consists of two MSAs (T-MSA, S-MSA) for the temporal and spatial axes in order, instead of a conventional single MSA block.
Specifically, the two MSAs are from pre-trained SwinIR and frozen, but a separate trainable adapter module is attached to the head of each MSA to adaptively process video data for spatial and temporal axes respectively.
Especially, trajectory-aware attention is applied to the T-MSA for better motion handling.
Finally, the method achieved a new SOTA performance in terms of PSNR and SSIM.

**Strengths:**

The main idea is simply to attach the trainable adapter modules to the fixed pre-trained SwinIR.
The proposed model is composed of existing modules, so it is easy to follow and re-implement, also the training is not that difficult.
It is a simple and practical solution.

**Weaknesses:**

There is no original module - The method is engineered from existing techniques without any modifications: SwinIR, HAT, Trajectory-aware attention, and Adapter. It can be a technical report but I think there is no academic contribution.
The proposed model applied HAT according to B5 in Table3, but this is not explained in the methodology section. It makes me feel the paper is incomplete.
It is weird the number of params of B1 in Table3 is 0M. I think only trainable params are counted, but the backbone's params should be also counted since it will occupy disk space (in ckpt file) and also GPU memory. Similarly, Table2 seems not fair because the proposed method does not include frozen params from SwinIR.

**Questions:**

The MSA from SwinIR is a kind of S-MSA, so I think it can be further improved to be used as a T-MSA but no effort is shown in the paper. Have the authors tried any experiments for this? Or is it sufficient?
The paper said "our method is generally applicable to different image pre-trained SR models", are there any results from different backbone networks other than SwinIR?

---

### Official Review · Reviewer_BWkb · 2023-10-28

**Soundness:** 4 excellent
**Presentation:** 4 excellent
**Contribution:** 2 fair
**Rating:** 6
**Confidence:** 3

**Summary:**

In this paper, a methodology (PFVSR) is introduced that involves training a Pre-trained image super resolution model and then Fine-tuning it for Video Super Resolution. The authors claim that this results in achieving performance comparable to or better than existing methodologies with less video data and a smaller model size. During the fine-tuning process, joint adaptation for spatial and temporal features was employed. The authors support their claim by comparing performance against various existing models across diverse datasets. Furthermore, the necessity of each component of the methodology is substantiated through an ablation study. Lastly, the paper demonstrates the model's performance through a range of qualitative results, including those in the appendix.

**Strengths:**

- In this paper, it is demonstrated that comparable or superior performance to existing Video Super Resolution Models can be achieved with significantly less video data. Considering that high-quality video data is more difficult to obtain than high-quality image data, such data efficiency is an advantage, and this performance is exhibited in Table 1.
- The paper presents an idea that allows for the training of a Video Super Resolution model utilizing both image and video data concurrently. In doing so, it secures model efficiency by fine-tuning a pre-trained Image Super Resolution model. This aspect can be verified through the comparison of model size, resource usage, and performance in Table 2.
- Thanks to the 2-stage strategy of first training the Image Super Resolution model and then fine-tuning the Video Super Resolution model, it is possible to conduct model debugging and performance evaluation more efficiently at various stages. Therefore, this methodology is expected to have high reproducibility and reliability.

**Weaknesses:**

- The parameters of the image super resolution model are being utilized in a frozen state for temporal adaptation, which is not straightforward. The role that the video super resolution model should perform in temporal adaptation is more closely related to Frame Interpolation. Moreover, there is no presented analysis or ablation study explaining why such a design choice was made.
- This paper differs from other compared baselines as it utilizes both image data and video data concurrently, hence the data used for training is significantly more abundant. Therefore, to ascertain whether the improved performance is due to the increased amount of data used, it is necessary to compare it with a unified model trained using the most naive approach. For example, one could train the model by repeating the image to match the frame of the video used in training.
- This methodology appears to be overly reliant on the image super resolution model, which could make it susceptible to chronic issues arising in Video Super Resolution. For example, problems occurring from sudden motions are difficult to resolve with the proposed temporal adaptation alone. [1]

**References**
[1] Tu, Z., Li, H., Xie, W., Liu, Y., Zhang, S., Li, B., & Yuan, J. (2022). Optical flow for video super-resolution: a survey. Artificial Intelligence Review, 55(8), 6505-6546.

**Questions:**

- Has there been any consideration or attempt to employ LoRa [2] for efficient temporal fine-tuning?
- What is the idea behind conducting the model efficiency experiments at low resolution as depicted in Table 2?
- Could you specify the number of frames that have been utilized during the training and inference phases?

**References**
[2] Hu, E. J., Shen, Y., Wallis, P., Allen-Zhu, Z., Li, Y., Wang, S., ... & Chen, W. (2021). Lora: Low-rank adaptation of large language models. arXiv preprint arXiv:2106.09685.

---

### Official Review · Reviewer_cshF · 2023-11-05

**Soundness:** 3 good
**Presentation:** 2 fair
**Contribution:** 1 poor
**Rating:** 1
**Confidence:** 5

**Summary:**

This paper introduces a novel framework called Pre-training and Fine-tuning Video Super-Resolution (PFVSR) for efficiently handling the video super-resolution (SR) task. The authors address the challenges of expensive computational resources, high dimensionality of video data, and scarcity of high-quality video SR datasets. They propose a two-step approach where a pre-trained image SR model is first fine-tuned for spatial adaptation and then adapted for temporal modeling. Through extensive experiments, the authors demonstrate that PFVSR achieves significant improvements in efficiency without compromising output quality compared to existing methods.

**Strengths:**

- It proposes a two-step approach that adapts pre-trained image SR models for spatial and temporal modeling, addressing the challenges in training video SR models.
- This paper utilizes pre-trained image SR models and adapts them for video SR tasks. It incorporates adapter modules for spatial and temporal adaptation, enhancing the model's capability to handle video sequences effectively.
- The paper conducts extensive experiments on public datasets to validate the proposed approach. It compares the performance of PFVSR with existing methods and showcases the significant improvements in efficiency without compromising output quality.

**Weaknesses:**

- Insights and analysis are rather limited.
    - This work simply reshapes the video patch embedding and finetunes the adapter without any investigation and analysis of the specific challenges of video SR. As it is believed that temporal propagation is the key element for video SR, the authors did not provide any analysis on why the temporal adaptation works and how well it works.
    - In Table 3, B1 + Spatial Adaptation (B2) is still an image-based model and generates video on a per-frame basis. Why simply adding an adapter can bring 1.1 dB PSNR gain?
    - In Table 3, B4 achieves 32.46 dB on REDS4, which is on par with BasicVSR++ and poorer than RSRT and RVRT. Compared to those SOTA methods, It seems the most performance gain stems from changing backbone (HAT), rather than the proposed Spatial-Temporal Adaptation.

- Technical contributions are limited.
    - The detailed architecture of Figure 1 looks almost the same as Figure 2 in AIM [1].
    - Section 3.3 is similar to Section 3.3 in [1].

- It is unclear how the authors adopt the trajectory-aware attention into the frozen MSA. The authors should provide more technical details if this is the default setting of the proposed PFVSR.
- In Section 4.1, the authors mentioned they initialize the SpyNet with pre-trained weights. However, there is no alignment module adopted for the adapter. Why SpyNet is used in implementation?

Overall the idea of repurposing pre-trained image SR models for video SR is interesting. However, the lack of significant analysis and unclear results in Table 3 could be misleading to the community. Given that, this paper is not publication-ready in its current form.

[1] AIM: Adapting Image Models for Efficient Video Action Recognition. ICLR'23

**Questions:**

See weaknesses

**Details Of Ethics Concerns:**

This paper exhibits significant overlap with [1], raising concerns about research integrity.

[1] AIM: Adapting Image Models for Efficient Video Action Recognition. ICLR'23